# Pressurized Liquid (PLE) Truffle Extracts Have Inhibitory Activity on Key Enzymes Related to Type 2 Diabetes (α-Glucosidase and α-Amylase)

**DOI:** 10.3390/foods12142724

**Published:** 2023-07-17

**Authors:** Eva Tejedor-Calvo, Diego Morales, Laura Morillo, Laura Vega, Mercedes Caro, Fhernanda Ribeiro Smiderle, Marcello Iacomini, Pedro Marco, Cristina Soler-Rivas

**Affiliations:** 1Department of Production and Characterization of Novel Foods, Institute of Food Science Research—CIAL (UAM + CSIC), C/Nicolas Cabrera 9, Campus de Cantoblanco, Universidad Autónoma de Madrid, 28049 Madrid, Spain; diego.morales@ucm.es (D.M.); laura.morillo@uam.es (L.M.); laura.vega@csic.es (L.V.); crisitna.soler@uam.es (C.S.-R.); 2Department of Forest Resources, Agrifood Research and Technology Centre of Aragon (CITA), Agrifood Institute of Aragón—IA2 (CITA-Zaragoza University), Av. Montañana 930, 50059 Zaragoza, Spain; pmarcomo@cita-aragon.es; 3AZTI, Food Research, Basque Research and Technology Alliance (BRTA), Parque Tecnológico de Bizkaia, Astondo Bidea, Edificio 609, 48160 Derio, Spain; mcaro@azti.es; 4Departmental Section of Galenic Pharmacy and Food Technology, Veterinary Faculty, Complutense University of Madrid, 28040 Madrid, Spain; 5Faculdades Pequeno Príncipe, Curitiba 80230-020, PR, Brazil; fhernanda.smiderle@professor.fpp.edu.br; 6Instituto de Pesquisa Pelé Pequeno Príncipe, Curitiba 80240-020, PR, Brazil; 7Department of Biochemistry and Molecular Biology, Federal University of Parana, Curitiba 81531-980, PR, Brazil; iacomini@ufpr.br

**Keywords:** truffles, β-D-glucans, ergosterol, fatty acids, antidiabetic, amylase, glucosidase

## Abstract

An optimized PLE method was applied to several truffle species using three different solvent mixtures to obtain bioactive enriched fractions. The pressurized water extracts contained mainly (1 → 3),(1 → 6)-β-D-glucans, chitins, and heteropolymers with galactose and mannose in their structures. The ethanol extracts included fatty acids and fungal sterols and others such as brassicasterol and stigmasterol, depending on the species. They also showed a different fatty acid lipid profile depending on the solvent utilized and species considered. Ethanol:water extracts showed interesting lipids and many phenolic compounds; however, no synergic extraction of compounds was noticed. Some of the truffle extracts were able to inhibit enzymes related to type 2 diabetes; pressurized water extracts mainly inhibited the α-amylase enzyme, while ethanolic extracts were more able to inhibit α-glucosidase. *Tuber brumale* var. *moschatum* and *T. aestivum* var. *uncinatum* extracts showed an IC_50_ of 29.22 mg/mL towards α-amylase and 7.93 mg/mL towards α-glucosidase. Thus, use of the PLE method allows o bioactive enriched fractions to be obtained from truffles with antidiabetic properties.

## 1. Introduction

Diabetes is one of the major problems in modern societies where unbalanced diets with a high sugar intake result in insulin resistance and the development of type 2 diabetes. This disease acts as a potent upstream event for many pathophysiological mechanisms such as oxidative stress, apoptosis, inflammation, and fibrosis [1]. The nutritional management of blood glucose imbalance has been a strategic target for many years, as there is mechanistic evidence suggesting that elevated blood glucose levels contribute towards the development of type 2 diabetes [2]. The key enzymes involved during digestion in glucose assimilation are α-amylase, which catalyzes the degradation of polysaccharides into oligosaccharides and disaccharides, and α-glucosidase, which transforms the latter into monosaccharides. Several natural extracts were tested as inhibitors of these enzymes, and among them, fungal extracts with hypoglucemic activities were described, i.e., an ergosterol extract from *Pleurotus ostreatus* was able to lower mouse blood glucose levels [3], and a *Inonotus obliquus* methanolic extract showed the highest enzyme inhibitory potential [4].

Together with mushrooms, truffles are also fungi. Those with the flavor for gourmet preparations belong to the Tuber genus; however, the genera *Terfezia* and *Tirmania* include odorless species referred as dessert truffles [5]. According to recent studies, the hypogeous fruiting bodies from both truffle types, in addition to their gastronomic interest, also showed biological properties that might be beneficial for human health [6,7].

Fungal compounds, such as β-glucans, chitins, and other polysaccharides, showed hypocholesterolemic, hypoglucemic, and immunomodulatory activities according to in vitro and in vivo tests [8,9,10]. Similarly, truffles polysaccharides were also able to reduce pro-inflammatory cytokines [6] and inhibited radical cell damage, acting as antioxidants [11]. Although truffles contain less than 8% fat, the lipophilic compounds were studied in detail because they play a key role in the aroma and flavor, as well as in their biological properties. For instance, fungal sterols were able to displace cholesterol from their dietary mixed micelles during digestion, acting as plant phytosterols [9,12], and they also have antitumoral, antidiabetic, and anti-inflammatory properties [13,14]. In addition to ergosterol, ergosta-7,22-dienol, and other derivatives characterized from mushrooms, truffles also contain brassicasterol, stigmasterol, or ergosta-5,8-dieno-3-ol [6,15], which show anti-inflammatory, antitumoral, and hypoglucemic bioactivities [16,17,18]. Other bioactive constituents present in truffles are phenolic compounds. Although their content and composition are different depending on reports, they seem to show antioxidant and immunomodulatory activities [19,20,21].

Environmentally friendly extraction technologies using pressurized liquids are widely used to obtain fractions containing different bioactive compounds from edible mushrooms and truffles [6,22]. Different molecules can be extracted depending on the solvent used, for instance, fungal polysaccharides were extracted using mainly pressurized hot water, yielding fractions with 20–90% truffle polysaccharides. The largest fungal sterols and lipid yields were obtained when 100% ethanol was used as the extraction solvent (20–30%) [10,22,23]. However, solvents combinations might improve the extraction of particular molecules with both lipophilic and hydrophilic characteristics that might show interesting bioactivities. In a previous study, a pressurized liquid extraction (PLE) method was optimized for two truffle species using water, ethanol, and a combination of both solvents [11], since the optimal conditions were independent of the truffle species utilized (180 °C and 16.7 MPa). In this work, the optimized PLE parameters were applied to seven truffles belonging to different genera in order to obtain fractions with hypoglucemic activities. The chemical composition of the obtained fractions was elucidated (carbohydrates, proteins, chitins, β-glucans, phenolic compounds, fatty acids, sterols) and their ability to inhibit the main enzymes involved in glucose absorption was studied in vitro.

## 2. Materials and Methods

### 2.1. Biological Material

Ascocarps from *Tuber aestivum* var. *uncinatum* (Chatin) I.R. Hall, P.K. Buchanan, Y. Wang and Cole, *Tuber borchii* Vittad., *Tuber brumale* var. *brumale* Vittad., and *Tuber brumale* var. *moschatum* (Bull.) I.R. Hall, P.K. Buchanan, Y. Wang and Cole were collected at Sarrión forests (Teruel, Spain). *T. indicum* Cooke and Massee, *T. magnatum* Pico, and *Mattirolomyces terfezioides* (Mattir.) E. Fisch. were supplied by Espora Gourmet company (Soria, Spain). Fresh truffles were identified, selected, and processed as indicated by Rivera et al. (2011) [24]. All truffle samples (moisture content approx. 80%) were lyophilized (Cryodos-50 Telstar, Barcelona, Spain), ground in a mill (Krups GVX 242, Zaragoza, Spain), and sieved until a particle size lower than 0.5 mm was obtained. Powdered truffles were kept at −80 °C until further use. PLE extracts obtained from *T. melanosporum, T. aestivum,* and *T. claveryi,* obtained as indicated by Tejedor-Calvo et al. (2023) [11], were used and compared with the other species as they were extracted with the same devices, solvents, and extraction conditions as the PLE extracts obtained in this work.

### 2.2. Reagents

Solvents including hexane (95%), chloroform (HPLC grade), methanol (HLPC grade), and acetonitrile (HPLC grade) were obtained from LAB-SCAN (Gliwice, Poland) and absolute ethanol, sodium carbonate (Na_2_CO_3_), and sulfuric acid (H_2_SO_4_) from Panreac (Barcelona, Spain). Potassium hydroxide (KOH), ascorbic acid, 2,6-Di-tert-butyl-*p*-cresol (BHT), bovine serum albumin (BSA), acetylacetone, p-dimethylaminebenzaldehyde, HCl (37%), phenol, hexadecane, ergosterol (95%), D-glucose, D-glucosamine hydrochloride, FAME (fatty acid methyl ester), and gallic acid were purchased from Sigma–Aldrich (Madrid, Spain) as well as acarbose and α-glucosidase from *Saccharomyces Cerevisae* and α-amylase from the porcine pancreas. All reagents and solvents were of analytical grade.

### 2.3. Pressurized Liquid Extractions (PLE)

According to the results obtained from the response surface plots and PCA analysis in a previous study [11], the combination of 180 °C and 16.7 MPa applied during 30 min were the optimal PLE conditions for *Tuber aestivum* and *Terfezia claveryi* truffles, although extractions from *Terfezia* truffles were more effective than from the *Tuber* genus. Thus, the optimized PLE method was applied to seven other truffles species.

Truffle powders (0.5 g) were submitted to pressurized liquid extraction using an Accelerated Solvent Extractor (ASE) (Dionex Corporation, ASE 350, USA) [6]. Briefly, samples were submitted to 16.7 MPa at 180 °C for 30 min. Fractions obtained with water were immediately frozen and freeze-dried in a LyoBeta 15 lyophilizer (Telstar, Madrid, Spain), and those obtained with ethanol were dried using a rotary vacuum evaporator at 40 °C (IKA^®^ RV 10, VWR International, Barcelona, Spain). The water:ethanol extracts were dried and then lyophilized. Afterwards, samples were stored in darkness at −20 °C.

### 2.4. Determination of Carbohydrates in PLE Extracts

Total carbohydrate content, β-glucan, and chitin levels of PLE extracts (50 mg/mL) were quantified as indicated by Tejedor-Calvo et al. (2021) [6] using D-glucose and glucosamine hydrochloride as standards for total carbohydrates and chitin determinations, respectively. Truffle polysaccharides were precipitated from truffle powder by adding ethanol (3:1 *v*/*v*). The mixtures were maintained at −20 °C overnight and centrifuged at 8000 rpm and 10 °C for 20 min. The obtained pellets were dialyzed (3.5 kDa Mr cut-off membrane Spectra/Por™ against water for 24 h at 4 °C) and subsequently freeze-dried. The monosaccharide composition of isolated polysaccharide fractions (1 mg) was analyzed by GC–MS, as described by Morales et al. (2018) [25]. The obtained alditol acetates were identified by their typical retention time and mass fragmentation to their corresponding monosaccharides, and compared to commercially available standards. Results were expressed as mol %, calculated according to Pettolino et al. (2012) [26]. Moreover, NMR spectra (^1^H, ^13^C and HSQC-DEPT) from isolated polysaccharide fractions were obtained using a 400 MHz Bruker model Advance III spectrometer with a 5 mm inverse probe. The samples (30 mg) were dissolved in Me_2_SO-*d*_6_ and analyzed at 70 °C. Chemical shifts were expressed in ppm (δ) relative to Me_2_SO-*d*_6_ at 39.7 (^13^C) and 2.40 (^1^H).

### 2.5. Determination of Lipid Compounds in PLE Extracts

Ergosterol and other unsaponifiable derivatives were quantified by GC-MS-FID, as described by Tejedor-Calvo et al. (2019) [27]. Ergosterol was used as standard and hexadecane (10% *v*/*v*) as internal standard for quantification.

The fatty acids analysis was carried out in a gas chromatograph 6890 N (Agilent Technologies, Santa Clara, CA, USA) with a flame ionization detector (FID) and capillary column HP-88 122-88A7 of 100 m, 0.25 mm i.d, and 0.2 μm film thickness from Agilent Technologies. Helium was used as carrier (3.5 mL/min) in the mode ’split injection’, and the injector and detector temperature were maintained at 250 °C and 290 °C, respectively. The oven temperature program was: 80 °C for 10 min, which was raised at 20 °C/min to 140 °C and 2 °C/min to 240 °C, maintained during 12 min, and then raised at 30 °C/min to 245 °C for 2 min to clean the column. Post-run conditions were 250 °C for three minutes. A mix of *cis*- and *trans*- fatty acid methyl esters (Supelco 37 component FAME mix, Merck Life Science S.L., Barcelona, Spain) (0.6 mg/mL) were used to identify the compounds by their retention times.

### 2.6. Determination of Protein and Phenolic Compounds

Soluble protein concentration (10 mg/mL) was evaluated using the Bradford method reagents (Sigma–Aldrich, Madrid, Spain) and total phenolic compound levels (10 mg/mL) by the Folin–Ciocalteu method, as indicated in a previous study [27]. BSA and gallic acid were used as standards for quantification.

### 2.7. Determination of the α-Amylase and α-Glucosidase Inhibitory Activities

PLE truffles extracts (10 mg) obtained with water were mixed with water up to 66.7 mg/mL. Extracts obtained by PLE using ethanol and ethanol:water (1:1 *v*/*v*) were adjusted to 25 mg/mL and solubilized in the same solvent mixtures. Samples were stirred in a vortex for 2 min, centrifuged at 12,000 rpm, and supernatants were used as the source of potential inhibitors. The extract ability to inhibit the key enzymes involved in carbohydrate digestion was determined by adapting the concentrations described in previous methods for α-glucosidase and for α-amylase to fit in the absorbance range and to get an effective reaction timing [28]. Briefly, truffle supernatants (10 µL) or acarbose (1 mg/mL) were mixed with 20 µL α-glucosidase in 100 mM sodium phosphate buffer pH 6.9 and incubated for 5 min. Afterwards, PNPG (2 mg/mL) was added (200 µL) and the reaction was spectrophotometrically followed (Genesys 10-S, Thermo Fisher scientific, Waltham, MA, USA) at 400 nm and 37 °C during 10 min. Similarly, truffle supernatants (100 µL) were mixed with 100 µL α-amylase (0.03 μL/mL 20 mM phosphate buffer pH 6.9 with sodium chloride 6 mM) and 100 µL starch 1% and then incubated at 20 °C for 3 min. Then, the mixture (200 μL) was mixed with DNS (100 µL) and heated at 100 °C for 10 min. One-hundred-microliters of this yellow–orange solution was diluted with 900 µL MilliQ water in a cuvette. The absorbance changes were measured at 540 nm. All assays were performed in duplicate. Acarbose and the most effective extracts were also assayed at different concentrations to stablish their IC_50_.

### 2.8. Statistical Analysis

Differences were evaluated at a 95% confidence level (*p*  ≤  0.05) using a one-way analysis of variance (ANOVA) followed by Tukey’s multiple comparison test.

## 3. Results

### 3.1. PLE Extractions from Several Truffle Species

Water was the more adequate solvent to extract major polysaccharides, proteins, and phenolic compounds and ethanol to obtain sterols enriched fractions. Although the mixture of ethanol:water (1:1) did not yield fractions of interesting bioactive compounds when *T. aestivum* or *T. claveryi* were studied, this mixture was also investigated because of the potential different response between species. The mixture could be a more selective medium to extract particular compounds (i.e., certain phenolic compounds) in any of the studied strains separating them from other highly hydrophilic or lipophilic compounds.

Extraction yields obtained using water (49–69%) were higher compared to the other solvents (40–49 and 12–20% from ethanol:water and ethanol, respectively). These extraction yields were lower than *T. melanosporum* extracts (64% using water as a solvent and 22% with ethanol) [6]. Similarly, *T. aestivum* and *T. claveryi* extracts yielded 69–71% using water, 22–32% using ethanol, and 45–49% when combining solvents [11]. When water PLE was used in mushrooms, lower yield extractions were obtained: 30.3 and 8% for *Pleurotus ostreatus* and *Ganoderma lucidum,* respectively. In comparison with microwave-assisted technology, the yield extraction was similar, but the PLE obtained a higher carbohydrate content in only one step [22]. *Mattirolomyces terfezoides* showed similar values using water or water:ethanol as extraction solvents (Table 1). The extractions carried out using the solvent mixture from *T. brumale* and *T. brumale* var. *moschatum* showed yields below 40% and those using ethanol from *T. brumale* and *T. magnatum* were below 15%.

These observations were in concordance with the observations reported before, suggesting that the harder texture of the *Tuber* genus might impair the extraction of water-soluble material more than *M. terfezoides* and *T. claveryi*, species belonging to the *Pezizaceae* family (with a softer hyphal wall). According to Tejedor et al. (2020), *T. brumale* var. *moschatum* contained a higher β-glucans content (31.1 g/100 g) than *T. magnatum* (10.4 g/100 g) and lower chitins (6.1 and 13.1 g/100 g, respectively) [6]. Therefore, the differences in extractabilities might not be because of the concentration of these two structural polysaccharides but because of their different binding or interactions with other compounds from the hyphal wall.

### 3.2. Carbohydrates Composition of PLE Fractions

Carbohydrates are the main constituents of truffle ascocarps (approx. 40% dw), and according to recent studies, most of them are β-glucans. In all the studied strains, more polysaccharides were extracted using pressurized water than any other solvent, but these effects were more notable for β-glucans extractions than for chitins. Indeed, the extracts obtained from some truffle species showed similar chitin values when pressurized water was used as solvent and when the solvent was water:ethanol (*M. terfezoides*, *T. borchii*, and *T. magnatum*) (Table 1). The TCH contents of the PLE extracts obtained with water were in the range of those recorded for *T. aestivum* and *T. claveryi* [11]. Within the selected truffle species, total β-glucan levels ranged between 10–31% dw, with the *Tuber brumale* var. *moschatum* and *T. indicum* varieties having the higher levels. Chitin concentrations were higher in *M. terfezoides* and *T. borchii*, ranging from 3.8 to 7.8% dw. However, the number of extracted β-glucans and chitins was significantly lower than *T. aestivum* and *T. claveryi* [11], suggesting that the noticed reduction in the extraction yield affected the extraction of these polysaccharides more than the chitins. The *M. terfezoides* water extract showed higher TCH levels than *T. claveryi*, but only 11.2 g/100 g β-glucans, while *T. claveryi* extracts contained 27.2 g/100 g, suggesting that the first strain might contain higher concentrations of other polysaccharides or lower molecular weight sugars. On the contrary, despite its low yield and lower TCH, the *T. brumale* extract contained more β-glucans than the *M. terfezoides* extract.

Fungal β-glucans were highlighted by several publications as the compounds responsible for the noticed hypoglycemic activities in vitro and in vivo [29,30]. Several mechanisms of action were proposed depending on the study, i.e., some authors indicated that they reduced intestinal blood glucose concentrations by delaying stomach emptying so that dietary glucose was absorbed gradually [30]. Other publications suggested a direct interaction with insulin receptors on target tissues [29]. Therefore, since the studies on truffle β-glucans are scarce, a more detailed study on their structure was carried out.

The monosaccharide composition of isolated truffle polysaccharides was species-dependent (although glucose was the major constituent in all the cases). *T. indicum* and *M. terfezoides* showed 8.6% galactose and *T. aestivum* var. *uncinatum* showed 5.4%, while the rest contained less than 1% (Table 2). Mannose levels were slightly higher in *T. indicum* and *T. magnatum* (31.8–33.5%) compared to the other strains. Glucose levels were reduced in those species with higher levels of the other two monosaccharides evaluated. According to previous works, *T. melanosporum* showed high glucose levels (75%), followed by mannose (24%) and low levels of galactose (0.8%) [6]. The high levels of glucose suggested the presence of a significant concentration of glucans, while the presence of mannose and galactose indicated that in those species, in addition to glucans, other heteropolysaccharides might be present, so their structure was also evaluated using NMR.

The NMR spectra of the polysaccharides from the different truffle species were obtained to investigate their anomericity and linkage type. However, only those obtained from a few species are shown in Figure 1 (those including larger spectral differences). All the studied species showed intense signals relative to the C-1 of β-D-Glc*p* (δ 102.5–102.7/4.17–4.43 ppm) as well as C-3 *O*-substituted (δ 85.9–86.1/3.17–3.39 ppm) signals, indicating the presence of (1 → 3)-β-D-glucan structures. Other signals confirmed CH_2_ *O*-substitution of the same units (inverted signals at δ 68.0/3.96 ppm and δ 68.0/3.49 ppm), suggesting (1 → 6)-linkages. However, the NMR spectra showed interferences and low intensity signals that might be from other cell wall molecules that were not completely disrupted. The existence of heteropolymers, including mannose and galactose, in their structure was confirmed by the signal at 100.7/4.87 relative to C-1 β-D-Man*p* [6]. However, only *T. aestivum* var. *uncinatum* showed a small signal.

Truffles from different species share similar (1 → 3) and (1 → 3),(1 → 6)-β-glucan structures but they might also contain different heteropolysaccharides showing particular monosaccharide compositions. Our results were in concordance with other studies showing (1 → 3),(1 → 6)-β-glucans in mushrooms [29]. The difference between species might be due to the cell wall polysaccharides being harder to extract, and consequently, their complete solubilization in the NMR solvent did not occur, hampering the signal detection.

### 3.3. Lipid Composition of PLE Fractions

In addition to β-glucans, lipid-containing extracts obtained from medicinal mushrooms were also suggested as potential fractions against diabetes. Fungal compounds such as oleic and linoleic acids together with sterols were suggested as potential contributors to the inhibitory activity reported on the key enzymes involved in glucose absorption [31].

Pressurized ethanol was a more suitable solvent than water or their combination to obtain more hydrophobic fractions from truffle species. Within the unsaponifiable lipids of PLE fractions, brassicasterol and ergosterol were the main sterols in all truffle extracts (Figure 1). Stigmasterol was only described for *T. aestivum* var. *uncinatum* and *T. magnatum* extracts [32,33], but it was also detected in the extracts obtained from other species such as *M. terfezoides* and *T. borchii* extracts. The last species, together with *T. aestivum* and *T. magnatum* truffles, are also called white truffles due to their white gleba, and because they are also phylogenetic, they are more related with each other than with the so-called black truffles [34]. Ergosta-7,22-dienol was noticed in almost all truffle extracts except for *T. brumale* var. *moschatum* and *T. indicum*. This derivative is also present in almost all mushroom species although in lower quantities than ergosterol [35].

The ethanol extract obtained from *T. aestivum* var. *uncinatum* contained larger amounts of sterols than the ethanol:water. However, since its extraction yield was lower, a similar sterol amount was extracted from the carpophore using ethanol:water (1.57 mg/g truffle) or only ethanol (1.59 mg/g truffle), suggesting that ethanol extraction was more selective. However, according to previous studies, *T. aestivum* var. *uncinatum* truffles contained 4.26 mg/g total sterols, indicating that approximately 40% of the sterols were extracted in this truffle [36]. Similar observations were noticed within the other truffles (ranging from 14 to 25%), suggesting that to increase the extracted sterol concentrations, other extraction parameters (i.e., extraction cycles) or more non-polar solvents (i.e., hexane) could be used, but in the latter case, the obtained PLE extract might not be of food-grade. According to previous studies, the saponifiable lipids from truffles included oleic (C18:1) and linoleic acids (C18:2) as the main compounds [37]. When the selected species were evaluated, both compounds together with linoleic acid (18:3) were detected as major fatty acid contributors in some truffle species such as *T indicum* or *T. borchii* (Figure 2).

The lipid profile of *M. terfezoides* was slightly different from those belonging to the *Tuber* genus. Some fatty acids, such as pentedecanoic acid (C15:0), cis-palmitoleic acid (C16:1), and arachidonic acid (C20:4w6), were detected in higher levels than the rest of the truffles. The use of pressurized ethanol extractions yielded fractions enriched in fatty acids with a profile different than the truffle fatty acid composition, and also than when ethanol:water was used. The latter solvent extracted more lower molecular weight fatty acids and also more nervonic acid (C24:1) but showed lowed lipid variability than the ethanol fraction. However, both solvents extracted other interesting minor fatty acids. Ethanol extracts contained, for instance, myristic acid (C14:0), cis-9-palmitoleic acid (C16:1), margaric acid (C17:0), and arachidonic acid (C20:4). Some of these fatty acids were also reported in *T. melanosporum*, *T. aestivum*, *T.borchii*, *Tirmania nivea*, and *T. maculatum* truffles [38]. Arachidonic acid was also previously found in traditional Chinese truffles from genus *Tuber*: *T. latisporum*, *T. subglobosum*, and *T. pseudohymalayense* [39].

### 3.4. Total Protein and Phenol Composition of PLE Fractions

Proteins and phenolic compounds were also evaluated in the obtained PLE extracts because according to a few reports, mushroom proteins (particularly lectins and lectin-like proteins) showed anti-diabetic properties in addition to other potentially beneficial effects [40]. Similarly, phenolic compounds might also be responsible for the inhibitory activity of methanol extracts on the main enzymes related to glucose metabolism [31].

Pressurized water extracted more proteins than the other tested solvents; however, the obtained concentrations were very low compared to the protein levels in truffles (approx. 9–20%) [6] (Table 1). The extraction using the ethanol:water mixture did not improve the protein extraction compared to water, the ethanol present in the extraction solvent might have induced their denaturation and precipitation, impairing their detection. Therefore, none of the pressurized liquids tested in this study may be appropriate to obtain protein enriched fractions from truffles. The combination of pressure with such a high temperature (180 °C) probably caused the breaking down of many of the free proteins or bound them to polysaccharides (i.e., by Maillard reactions), as suggested elsewhere [11].

Phenolic compounds are molecules that are usually were found in low quantities in truffles, i.e., the *T. brumale* var. *moschatum* content was 1.13 mg/g while *T. melanosporum* ranged from 1.09 to 2.61 mg/g [6]. The highest total phenolic compound (TPC) values within the PLE fractions were obtained using water as a solvent followed by ethanol:water and the lowest was with ethanol, indicating that most of the truffle phenolic compounds were polar molecules. However, a wide number of phenolic compounds were described in mushrooms and truffles showing a very different chemical nature [41,42,43]. Therefore, the phenolic compounds present in the extracts obtained with different solvents might be completely different depending on polarities. *M. terfezoides* and *T. borchii* aqueous and ethanol:water extracts contained more TPC than the rest of the species compounds. However, the extract obtained using ethanol extracted more TPCs from *T. magnatum* than from the other species, suggesting a different phenolic composition (with different polarities) among the selected species. Considering the obtained yield, PLE extractions can be considered as an interesting tool to obtain highly concentrated fractions of phenolic compounds. For *T. aestitvum* var. *uncinatum*, for instance, 1.54 mg/g extract were obtained, and this species showed 1.09 mg/g truffle; therefore, almost 83% of the TPCs were obtained from the ascocarp. The TPC yields for the other species ranged from 75 to 80%.

### 3.5. PLE Extract Inhibitory Activity on the Key Enzymes Linked to Diabetes

Inhibition of pancreatic α-amylase will induce a reduction in starch hydrolysis and inhibition of intestinal α-glucosidase will limit glucose absorption; therefore, reducing the activity of both enzymes might be a potential strategy to prevent type 2 diabetes. Some inhibitors, such as acarbose, are used to prevent diabetes or other metabolic disorders [44]. Therefore, the ability of the PLE extracts obtained from truffles to inhibit their activity was tested.

When the extracts obtained with pressurized water were tested (66.7 mg/mL), the results indicated that only those obtained from *M. terfezoides* showed a high α-amylase inhibitory capacity. It showed levels similar to other species, such as *T. claveryi* and *T. aestivum*, suggested in other previous studies as species with interesting inhibitory activity of both enzymes [11] (Figure 3A). Values near 100% of inhibition indicate that the enzyme is not active. As noticed for the other desert truffle, *M. terfezoides* was less effective at reducing the activity of α-glucosidase. Water extracts from *T. brumale* var. *moschatum* inhibited the activity of α-amylase more than α-glucosidase, while *T. indicum* inhibited the α-glucosidase more. However, both water extracts were less effective than *M. terfezoides*. When ethanol and ethanol:water extracts were applied at similar concentrations to the water extracts, they all completely inhibited the enzymes; therefore, they were diluted and tested at 25 mg/mL. Then, the results indicated that almost all of the extracts obtained from the selected truffle species using ethanol were able to inhibit more than 50% of the activity of α-glucosidase at levels similar to 1 mg/mL acarbose (Figure 3B).

However, only few of them managed to inhibit the activity of α-amylase more than 50%. *M. terfezoides* and *T. brumale* var. *moschatum* showed a higher α-amylase inhibitory capacity than α-glucosidase; however, *T. aestivum* var. *uncinatum* and *T. brumale* were more effective at inhibiting α-glucosidase. Surprisingly, the inhibitory activities of the ethanol:water extracts were different than those of ethanol; in most of the species, they were more effective at inhibiting the α-amylase than the α-glucosidase (Figure 3C). For the same species, their ethanol extracts were more effective at inhibiting α-amylase (i.e., *T. indicum*, *T. magnatum*, *T. borchii*, etc.), suggesting that the inhibitory compounds for both enzymes might be of a different nature, with α-amylase inhibitors being more polar than those acting against α-glucosidase. A similar effect was noticed with the PLE extracts of other truffle species, such as *T. melanosporum,* obtained under the same conditions [6]. The inhibitors presence was not species dependent because, for instance, the water and ethanol:water extracts from closely related varieties, such as *T. brumale* and *T. brumale* var. *moschatum*, showed similar inhibitory responses towards α-amylase and α-glucosidase, with the inhibitory activity of α-amylase always being higher. However, the ethanol extracts showed a completely different behavior. Similarly, the ethanol:water and ethanol extracts from *T. aestivum* and *T. aestivum* var. *uncinatum* showed similar inhibitory responses towards both enzymes, with the inhibitory activity of α-glucosidase being higher. Moreover, the water extract of *T. aestivum* was very effective but its variety *uncinatum* showed almost no inhibition at all for any of the enzymes.

When the IC_50_ levels of the PLE extracts showing interesting inhibitory activities for both enzymes were calculated, the results indicated that ethanol extracts showed lower levels than water extracts (Table 3), but there was not one species that could similarly inhibit both enzymes with low IC_50_s. The PLE extract obtained using water as a solvent from *M. terfezoides* showed a lower IC_50_ for both enzymes than other desert truffles, such as *T. claveryi* [11], but higher than for other Tuber species, such as *T. aestivum*. According to a previous report, polysaccharides might be the responsible compounds for the noticed enzyme inhibition [45,46]. *M. terfezoides* amylase inhibitory activity might be related to the large TCH levels, as well as the high galactose content in its polysaccharides, indicating the presence of particular heteropolysaccharides, in addition to β-glucans.

The IC_50_ values for the ethanol extracts were in the range of those found for other species (*T. aestivum* and *T. claveryi*) [11], but they were higher than in edible mushrooms, such as *Agaricus blazei*, *Coprinus comatus*, and *Morchella conica*, with IC_50_ values of 0.71–1.72 mg/mL [4]. Fatty acids [47,48], as well as phenolic compounds or polyphenols [49], might be the responsible for these inhibitions. The fatty acids might be more involved in the α-glucosidase inhibition as they might be present in higher concentrations in the more apolar extracts, such as those with only ethanol. Although ethanol:water extracts showed higher phenol concentrations, the α-amylase inhibition was, for many species, larger than in that obtained with pressurized ethanol. TPC were higher in the *M. terfezoides* ethanol extract, which was the extract showing higher α-amylase inhibitory activity. Thus, no clear correlation between molecule types could be established and further studies are needed to try to verify whether the noticed inhibitory activities are due to the presence of one or a few particular compounds.

## 4. Conclusions

The application of pressurized liquid as an extraction method allowed fractions with potential antidiabetic activity to be obtained. Pressurized water extracts contained mainly (1 → 3)-β-D-glucans, as well as (1 → 3),(1 → 6)-β-D-glucans, depending on the considered strain. They also contained chitins and heteropolymers, including galactose and mannose, in their structure. However, sterols and fatty acids were mainly present in ethanolic extracts. Both types of PLE extracts where able to inhibit the two key enzymes related to type 2 diabetes. The combination of both solvents yielded extracts with an intermedia composition that showed no synergy; however, they extracted more compounds able to inhibit α-amylase while ethanol extracts contained more compounds able to inhibit α-glucosidase. The ethanol extracts from *Mattyrolomyces terfezoides* and *T. brumale* var. *moschatum* showed the lowest IC_50_ against α-amylase but *T. aestivum* var. *uncinatum* and *T. brumale* showed the lowest IC_50_ against α-glucosidase. These results indicate that environmentally friendly extraction methods can be used to produce enriched natural extracts from truffles. Further studies incorporating these extracts into bioactive foods are needed to explore their stability.

## Figures and Tables

**Figure 1 foods-12-02724-f001:**
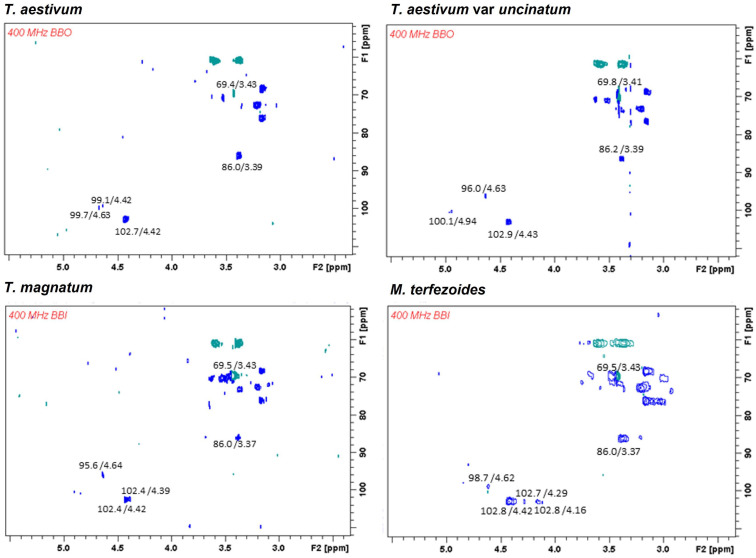
HSQC spectra of the isolated polysaccharides from *T. aestivum*, *T. aestivum* var. *uncinatum*, *T. magnatum*, and *M. terfezoides*. Experiments were performed in Me_2_SO-*d*_6_ at 70 °C (chemical shifts are expressed in δ ppm).

**Figure 2 foods-12-02724-f002:**
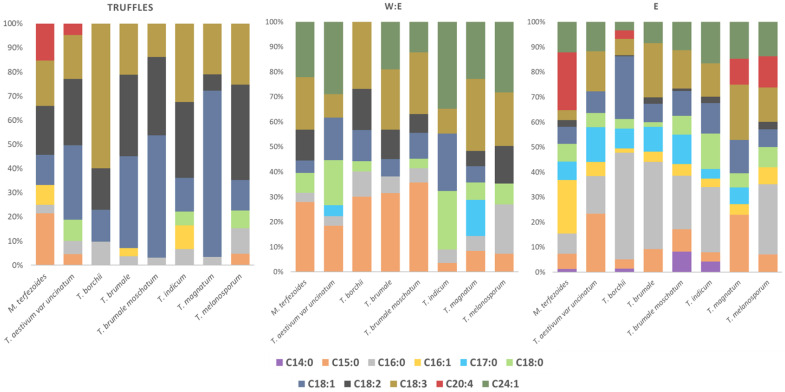
The relative percentages of the identified fatty acids in truffle ascocarps and PLE fractions obtained with ethanol (E) and ethanol:water (1:1) (E:W).

**Figure 3 foods-12-02724-f003:**
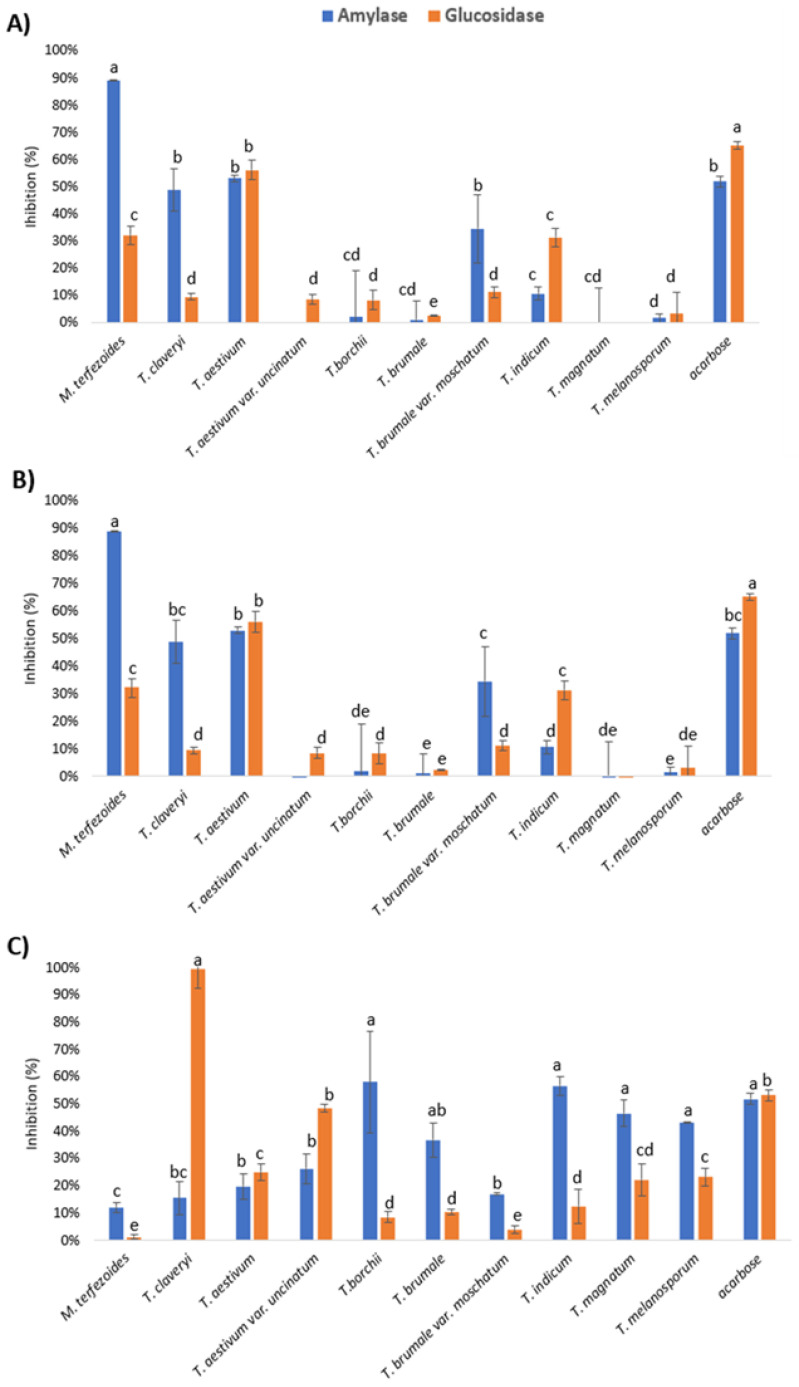
Percentage inhibition of α-amylase (blue) and α-glucosidase (orange) from PLE extracts obtained using pressurized (**A**) water (66.7 mg/mL), (**B**) ethanol (25 mg/mL), and (**C**) water:ethanol (25 mg/mL). Different letters denote significant differences (*p* ≤ 0.05) between different truffle species (*p* ≤ 0.05).

**Table 1 foods-12-02724-t001:** Yields and concentrations of the main compounds extracted by PLE during 30 min at 180 °C and 16.7 MPa from several truffle species. Indicated values are *w*/*w* extract (-: not detected). Data from TCH, β-glucans, chitin, and proteins are shown in g/100 g extract. Data from total sterols, minor sterols, and TPC are shown in mg/g extract.

Species	Yield %	TCH	β-Glucans	Chitin	Soluble Proteins	Total Sterols	Ergosterol	Brassicasterol	Ergosta-7,22-Dienol	Stigmasterol	TPC
W											
*M. terfezoides*	69.62 ± 5.32	46.39 ± 1.29	16.12 ± 1.98	8.83 ± 0.23	7.15 ± 0.85	-	-	-	-	-	2.77 ± 0.07
*T. aestivum* var. *uncinatum*	58.72 ± 3.31	38.01 ± 2.56	16.47 ± 1.03	11.21 ± 1.02	3.83 ± 0.35	-	-	-	-	-	1.54 ± 0.12
*T. borchii*	56.75 ± 2.87	37.30 ± 1.98	15.82 ± 2.01	12.16 ± 1.14	7.85 ± 0.70	-	-	-	-	-	2.07 ± 0.02
*T. brumale*	48.51 ± 3.02	37.37 ± 2.07	26.20 ± 1.51	11.36 ± 0.99	4.58 ± 0.28	-	-	-	-	-	1.52 ± 0.14
*T. brumale* var. *moschatum*	47.30 ± 2.09	29.43 ± 3.01	14.61 ± 0.57	10.93 ± 0.12	6.44 ± 0.13	-	-	-	-	-	1.36 ± 0.10
*T. indicum*	49.62 ± 4.47	35.51 ± 0.87	18.88 ± 2.12	12.54 ± 0.67	4.35 ± 0.82	-	-	-	-	-	1.93 ± 0.08
*T. magnatum*	49.46 ± 4.67	38.60 ± 1.64	12.74 ± 1.22	10.01 ± 0.43	4.56 ± 0.86	-	-	-	-	-	1.48 ± 0.13
W:E											
*M. terfezoides*	49.41 ± 3.45	21.99 ± 0.87	5.22 ± 1.09	8.62 ± 0.65	2.26 ± 0.35	2.06 ± 0.15	0.61 ± 0.01	0.85 ± 0.01	0.61 ± 0.01	-	1.99 ± 0.12
*T. aestivum* var. *uncinatum*	40.25 ± 1.47	12.55 ± 1.13	4.10 ± 0.67	10.41 ± 1.10	1.84 ± 0.41	3.90 ± 0.83	1.52 ± 0.12	1.71 ± 0.43	-	0.65 ± 0.08	1.26 ± 0.20
*T. borchii*	40.43 ± 2.16	16.18 ± 2.25	3.19 ± 0.34	12.91 ± 3.17	2.41 ± 0.15	2.00 ± 0.14	0.76 ± 0.01	0.74 ± 0.05	0.52 ± 0.01	-	1.62 ± 0.07
*T. brumale*	34.07 ± 1.68	9.24 ± 1.65	2.08 ± 0.15	6.55 ± 1.06	1.68 ± 0.17	1.82 ± 0.56	0.35 ± 0.10	0.99 ± 0.09	0.50 ± 0.23	-	1.40 ± 0.14
*T. brumale* var. *moschatum*	37.78 ± 1.78	9.96 ± 1.89	2.30 ± 0.87	7.07 ± 0.82	3.01 ± 0.20	1.48 ± 0.38	0.52 ± 0.10	0.95 ± 0.11	-	-	1.26 ± 0.17
*T. indicum*	42.80 ± 2.46	13.49 ± 2.09	3.04 ± 1.12	10.26 ± 1.33	2.27 ± 0.09	2.17 ± 0.56	1.10 ± 0.24	1.08 ± 0.22	-	-	1.41 ± 0.10
*T. magnatum*	40.92 ± 3.55	15.91 ± 2.16	3.64 ± 0.93	10.97 ± 1.27	3.42 ± 0.39	3.64 ± 0.74	0.92 ± 0.23	0.93 ± 0.23	0.90 ± 0.12	0.90 ± 0.07	1.15 ± 0.09
E											
*M. terfezoides*	19.85 ± 3.21	-	-	-	-	3.43 ± 0.35	0.82 ± 0.10	0.93 ± 0.08	0.91 ± 0.01	0.76 ± 0.09	0.68 ± 0.07
*T. aestivum* var. *uncinatum*	17.08 ± 2.87	-	-	-	-	9.31 ± 0.65	2.60 ± 0.12	2.66 ± 0.16	2.58 ± 0.27	1.46 ± 0.08	0.54 ± 0.01
*T. borchii*	20.78 ± 2.69	-	-	-	-	4.04 ± 0.41	1.13 ± 0.18	1.10 ± 0.10	0.72 ± 0.01	1.06 ± 0.19	0.55 ± 0.03
*T. brumale*	12.99 ± 1.15	-	-	-	-	6.00 ± 0.64	2.52 ± 0.02	2.60 ± 0.38	0.85 ± 0.12	-	0.60 ± 0.12
*T. brumale* var. *moschatum*	16.77 ± 1.90	-	-	-	-	4.17 ± 0.70	2.43 ± 0.24	1.75 ± 0.37	-	-	0.68 ± 0.09
*T. indicum*	18.86 ± 2.01	-	-	-	-	2.92 ± 0.45	1.53 ± 0.11	1.39 ± 0.24	-	-	0.70 ± 0.10
*T. magnatum*	14.76 ± 3.00	-	-	-	-	7.86 ± 0.54	2.31 ± 0.02	2.34 ± 0.02	2.30 ± 0.27	0.88 ± 0.12	0.77 ± 0.11

**Table 2 foods-12-02724-t002:** Monosaccharide composition of the polysaccharides isolated from the different truffle species.

Species	Mannose (%)	Glucose (%)	Galactose (%)
*M. terfezoides*	15.1	76.3	8.6
*T. aestivum* var. *uncinatum*	26.0	68.6	5.4
*T. borchii*	20.3	78.8	0.9
*T. brumale*	16.1	83.5	0.4
*T. brumale* var. *moschatum*	27.8	70.2	2.0
*T. indicum*	31.8	59.6	8.6
*T. magnatum*	33.5	65.8	0.7

**Table 3 foods-12-02724-t003:** Glucosidase and amylase IC_50_ indexes of the truffle PLE extracts showing higher inhibitory activity for both enzymes.

	α-Amylase IC_50_ (mg/mL)	α-Glucosidase IC_50_ (mg/mL)
Water		
*M. terfezoides*	29.22	119.43
Ethanol		
*M. terfezoides*	6.03	23.0
*T. aestivum* var. *uncinatum*	19.95	7.93
*T. brumale*	26.22	9.19
*T. brumale* var. *moschatum*	9.39	28.33
Arcabosa (1 mg/mL)	0.67	0.83

## Data Availability

Data is contained within the article.

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
