# Peer review of "Pressurized Liquid (PLE) Truffle Extracts Have Inhibitory Activity on Key Enzymes Related to Type 2 Diabetes (α-Glucosidase and α-Amylase)"

_foods, 2023, doi:10.3390/foods12142724_

Round 1

Reviewer 1 Report

This paper showed the characteristics of extracts prepared using the PLE method for several truffle and the α-glucosidase and α -amylase inhibitory effects of the extracts.  The PLE method has received a lot of attention and is noteworthy. However, there are several issues that should be revised for acceptance.

To discussion the characteristics of extracts prepared by the PLE method, a comparison with extracts prepared by other methods should be necessary.

In the inhibitory activity of those enzymes, the water and ethanol extracts showed the same activity (Figure 3), even though the components are completely different. The reasons for them should be fully discussed.

Although the ethanol and ethanol-water extracts are not water soluble, it is a wonder that the enzyme activity has been properly evaluated.

Each peak of NMR spectra should be assignment.

The definition of 100% of inhibitory activity should be indicated.

 Minor editing of English language required

Author Response

Point 1: This paper showed the characteristics of extracts prepared using the PLE method for several truffle and the α-glucosidase and α -amylase inhibitory effects of the extracts.  The PLE method has received a lot of attention and is noteworthy. However, there are several issues that should be revised for acceptance.

Response 1: Thank you for your critical comments of the work. We appreciate your comments, and we believe they will help us to increase the quality of this publication. Changes have been made throughout the manuscript. Please check the modifications made with track changes.

Point 2: To discussion the characteristics of extracts prepared by the PLE method, a comparison with extracts prepared by other methods should be necessary.

Response 2: Thank you for your advice. In the 3.1 section, the enriched fractions composition was compared with other enriched extracts from other truffles species obtained with PLE method. Apart of PLE, SFE has been used to extract lipid compounds in truffle samples. But conventional methodologies, such as microwave or ultrasounds have not been applied in truffles yet. However, we have included the comparison with microwave and ultrasound extraction technology in mushrooms.

Point 3: In the inhibitory activity of those enzymes, the water and ethanol extracts showed the same activity (Figure 3), even though the components are completely different. The reasons for them should be fully discussed.

Response 3: This information is later discussed with the Table 3 results.

Point 4: Although the ethanol and ethanol-water extracts are not water soluble, it is a wonder that the enzyme activity has been properly evaluated.

Response 4: In the 2.7 section is explained the methodology used: ‘Extracts obtained by PLE using ethanol and ethanol:water (1:1 v/v) were adjusted to 25 mg/ml and solubilized in the same solvent mixtures.’. Therefore, the extracts were previously solved in the corresponding solvent. 

Point 5: Each peak of NMR spectra should be assignment.

Response 5: Regarding the β-D-glucan, we were only interested in signals indicating anomericity and linkage type: δ 102.5–102.7/4.17–4.43 ppm (C-1 of β-D-Glcp), δ 85.9–86.1/3.17–3.39 ppm (C-3 O-substituted), δ 100.7/4.87 (C-1 β-D-Manp) as well as δ 68.0/3.96 ppm and δ 68.0/3.49 ppm (CH2 O-substitution). Therefore, only the most interesting signals related with the polysaccharides were marked in the Figure 3 and discussed in the manuscript.

Point 6: The definition of 100% of inhibitory activity should be indicated.

Response 6: The 100% of the enzyme activity correspond to 0% of the enzyme inhibition. Therefore, the 100% of the enzyme inhibition indicate that the enzyme is not active. We have indicate it in the results section.

Reviewer 2 Report

The comments are as follows:

1. Please, improve the abstract by inserting the overall conclusion.

2. Please, add moisture content of the sample material to the section 2.1.

3. The text from line 180-184 should be moved to the section 2.3.

4. The authors should include SD values in Table 1.

5. The practical applications of the findings has to be expressed more. What are the future applications? What are the next research directions? Please, improve the Conclusions section.

6. Are really necessary the eight self-citations by dr. Diego Morales and twelve self-citations by dr. Cristina Soler-Rivas? Please, revise it.

7. Please, revise the text for misprints.

Author Response

Point 1: Please, improve the abstract by inserting the overall conclusion.

Response 1: A conclusion in the abstract has been included.

Point 2: Please, add moisture content of the sample material to the section 2.1.

Response 2: The moisture percentage in the sample was approx. 80%, but after the application of freeze-dried technique the moisture was 0 %. Anyway, we have included the moisture content in the 2.1 section of the manuscript.

Point 3: The text from line 180-184 should be moved to the section 2.3.

Response 3: The paragraph has been removed to materials and method section.

Point 4: The authors should include SD values in Table 1.

Response 4: The Table 1 has been modified.

Point 5: The practical applications of the findings has to be expressed more. What are the future applications? What are the next research directions? Please, improve the Conclusions section.

Response 5: Thank you for your comment. A new paragraph has been added in the results and discussion section. Also, conclusions have been modified.

Point 6: Are really necessary the eight self-citations by dr. Diego Morales and twelve self-citations by dr. Cristina Soler-Rivas? Please, revise it.

Response 6: Sorry, we only check the number of self-citations of the first author (6 publications). In all of them, the research Morales, D. and Soler-Rivas, C. are co-authors. We have carefully checked the other publications of these authors and we have removed those publications that we can dispense with. Some of them refers to methods used in this manuscript, and we believe that the rest are necessary to make a proper discussion.

Point 7: Please, revise the text for misprints.

Response 7: Thank you for your advice. The manuscript has been reviewed.
